# TEXT TO STEALTHY ADVERSARIAL FACE MASKS

## ABSTRACT

Recent studies have demonstrated that modern facial recognition systems, which are based on deep neural networks, are vulnerable to adversarial attacks, including the use of accessories, makeup patterns, or precision lighting. However, developing attacks that are both robust (resilient to changes in viewing angles and environmental conditions) and stealthy (do not attract suspicion by, for example, incorporating obvious facial features) remains a significant challenge. In this context, we introduce a novel diffusion-based method (DAFR) capable of generating robust and stealthy face masks for dodging recognition systems (where the system fails to identify the attacker). Specifically our approach is capable of producing high-fidelity printable textures using the guidance of textual prompts to determine the style. This method can also be adapted for impersonation purposes, where the system misidentifies the attacker as a specific other individual. Finally, we address a gap in the existing literature by presenting a comprehensive benchmark (FAAB) for evaluating adversarial accessories in three dimensions, assessing their robustness and stealthiness.

## 1 INTRODUCTION

Facial recognition systems have increasing prominence, with applications in a range of environments. Importantly, these systems aim to accurately classify an individual when presented with an image of them, hence, adversarial attacks against such systems are important to identify and explore. Deep learning facial recognition systems, the state of the art technique for biometric identification (Vakhshiteh et al., 2021), have a history of said attacks, causing the systems to behave in an unintended manner when presented with images that have been carefully modified by attacks.

Previous studies on the matter have used a plethora of both attack surfaces and techniques to misdirect these systems into misclassifying individuals. Some explore attacks by digitally perturbing images of faces (Lin et al., 2023), whilst others use makeup (Yin et al., 2021; Sun et al., 2024) or accessories (Sharif et al., 2019; Komkov & Petiushko, 2021; Zolfi et al., 2022; Gong et al., 2024; Pautov et al., 2019; Xiao et al., 2021). Traditionally, gradient descent based approaches have been employed to generate accessories, to much success (Zolfi et al., 2022); however, whilst these achieve robustness to changes in viewing angles and environmental conditions, they lack in stealthiness – the need for the attacks to be undetectable by human observers.

Many developments have been made to this regard in order to balance the adversarial strength of an attack with the style and realism of the perturbations. Various loss functions have been explored such as total variation loss (Mahendran & Vedaldi, 2015) which makes the perturbations smoother, making an attack more stable, realistic and robust to interpolation techniques (Komkov & Petiushko, 2021; Zolfi et al., 2022). Other work has used style extractors, L1 losses with a reference style, to make a style adapt to an attack in order to encourage the generation of a stealthy accessory that would not raise suspicion in the real world (Gong et al., 2024). A common struggle with these approaches is generating perturbations that look stealthy consistently, especially against larger facial recognition models such as those based on ResNet (He et al., 2016). When attacks are not attempting to maximize stealthiness, the final perturbations often contain facial features and noise-like perturbations. On the other hand, when attacks prioritize stealthiness, their efficacy is significantly reduced.

Recent literature for general adversarial attacks have propagated towards the use of generative models to support the generation of realistic adversarial examples and perturbations. These methods use a pretrained model to produce or manipulate an adversarial sample towards a given style. Song

et al. (2018) used generative adversarial networks (GANs) to generate significantly more realistic examples than were possible with perturbation based methods. Alternatively, diffusion models have too been shown to support generation of adversarial samples (Xue et al., 2023; Chen et al., 2023; Dai et al., 2023) and have several desirable properties for this task, such as greater interpretability, controllability and visual fidelity in the produced samples (Dai et al., 2023).

Diffusion models have been used to generate adversarial makeup (Sun et al., 2024), but to the best of our knowledge, there has been no work on their use in the creation of adversarial accessories. Since the COVID-19 pandemic, the use of face masks by the general public has increased and makes them a prime adversarial accessory surface as they cover a substantial area of the face (Zolfi et al., 2022; Gong et al., 2024). By using adversarial guidance (Dai et al., 2023) during the generative process, and text prompts

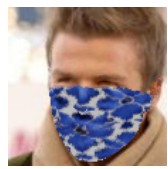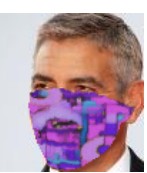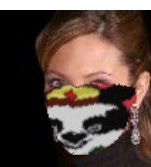

Figure 1: Adversarial DAFR masks against Mobile-FaceNet for "David Beckham", "George Clooney" "Angelina Jolie" from the PubFig dataset (Kumar et al., 2009).

to control the style, adversarial optimization and style generation can happen simultaneously, allowing the adversarial perturbations to become part of the style content and leading to truly stealthy and robust adversarial face masks. To this end, we propose the resulting diffusion-based face mask attack that we call Diffusion Attack against Facial Recognition (DAFR) that is able to achieve state of the art stealthiness in a white-box threat model, where the attacker has access to the victim model's weights. In addition to a new novel attack, we propose a benchmark to tackle the current inconsistent experimental frameworks and results within the field, largely caused by varying threat models and attack objectives. This system, titled the Face Accessory Attack Benchmark (FAAB), has been designed with flexibility at its core, allowing it to be adapted to a wide range of attack objectives, so that more consistent evaluation and comparison of attack methods can be performed, focusing on robustness to different conditions, stealthiness and adversarial strength.

In summary, our main contributions are:

- A novel diffusion-based stealthy adversarial face mask generation method, titled DAFR, which uses adversarial guidance to produce adversarial textures that retain the content of the reference images and that can be styled using text prompts. The resulting generated face masks are stealthy, robust to environmental changes, and comparable to previous work.

- A robust benchmarking framework, called FAAB, that includes a set of standardized tests and procedures to evaluate the performance of accessories. The framework supports frequently used statistics like cosine distances, success rates, and a new metric that we discuss later that is based on CMMD, in order to evaluate the stealthiness of generated textures quantitatively. In addition, the modular design of the benchmark allows each component to be easily interchanged in order to suit each attack's objective.

## 2 DAFR: DIFFUSION ATTACK AGAINST FACIAL RECOGNITION

**Facial Recognition:** Modern facial recognition networks are often Siamese networks (Bromley et al., 1993) that are designed to work with a large number of classes and with potentially unseen identities during testing (Wen et al., 2016). These models can be split into two components: the backbone and head. The backbone takes in an image and outputs the embedding of that image in the learnt feature space, which can then be fed into a head for final classification. The embedding spaces are trained to be discriminative and to be effective for multiple different heads for different recognition problems. Recent adversarial work focuses on attacking the backbone directly rather than the head (Vakhshiteh et al., 2021; Zolfi et al., 2022; Gong et al., 2024) and we follow suit. A further discussion of facial recognition systems can be found in appendix B.

**DAFR:** The objective of DAFR is to produce textures for face masks that are not only adversarial, but stealthy as well. To do this, the reverse diffusion process of a diffusion model can be manipulated such that the final output, $m_a$, looks like the output if the reverse process was not manipulated (i.e, is stealthy) and is adversarial. For a texture to be adversarial, it must have a low cosine similarity with the anchor embedding, $e_a$, of the attacker and preferably be under a recognition threshold such that

the network would not recognize the masked image as the attacker, this is called *dodging*. If the similarity is maximized with the anchor of a specific other identity, then it is called *impersonation*. We focus on dodging, but our attack can be adapted for impersonation too.

It is well documented that adversarial patches (and thus adversarial accessories too) must consider different real world transformations during generation so that the resulting accessories would exhibit robustness to these transformations when they do occur (Athalye et al., 2018). To achieve this, we generate our face masks using a set of images of the attacker, $H$, and optimize over these. Moreover, since face masks are 3D objects, to generate physically realizable 3D masks, the generated textures must be rendered onto the face during generation so the generation conditions match the real world.

One way to control the generation of a diffusion model is classifier guidance where the scores (the gradient of the log of a function) of a classifier are used during generation to perform conditional generation (Dhariwal & Nichol, 2021). AdvDiff is a recent diffusion-based adversarial attack that uses adversarial guidance (Dai et al., 2023), based on classifier guidance, to control the generation of a class conditional latent diffusion model (LDM, Rombach et al. (2022)) to generate unrestricted adversarial examples for ImageNet (Deng et al., 2009). We fuse together adversarial guidance and 3D rendering to allow for a more advanced procedure to generate samples that can act as textures for stealthy adversarial masks, as demonstrated in algorithm 1. Since we use LDM's, adversarial guidance is applied to latents, not on the pixel level. $f$ is defined in equation (1).

---

**Algorithm 1:** Diffusion Attack on Facial Recognition (DAFR)

---

**Input:** set of attacker pictures ($H$), text prompt ($c$), dodging sign ($d$), anchor embedding ($e_a$), adversarial limit ($l$), iterations of the adversarial loop ($k$), adversarial guidance weight ($s$), facial recognition backbone ($E$), generation timesteps ($T$)

$x_T \sim \mathcal{N}(0, \mathbf{I})$
**for** $t$ from $T$ to *1* **do**
    Sample $x_{t-1}$ using classifier free sampling, using $x_t$ and $c$
    **if** $t/T \leq l$ **then**
        **for** $i$ from *1* to $k$ **do**
            $h_n$ = Next image in $H$
            *// d should be -1 if dodging and +1 if impersonating*
            $x_{t-1} = x_{t-1} + ds \cdot f(t/T) \cdot \nabla_{x_{t-1}} \cos(E(R(x_{t-1}, h_n)), e_a)$

**return** $x_0$

---

One of the first challenges faced when designing DAFR was adding the flexibility in style desired for a stealthy mask. For this, we chose to use text-to-image models, rather than class conditional models, to condition the generation of samples such that the style is dictated by a text prompt. Classifier free sampling (Ho & Salimans, 2022) is the dominant method for conditioning generation and can be used in conjunction with the guidance, allowing for the generation to achieve both goals.

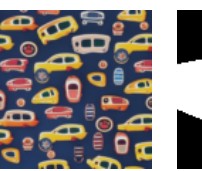 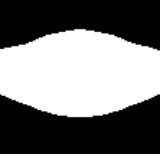 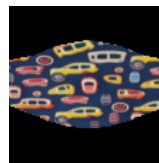

Figure 2: The left image is a generated texture, the middle is the UV mask and the right is a processed mask texture. We refer to the leftmost image as the texture image and a cropped version of the rightmost image (figure 4) as the masked texture image.

Additionally, we needed to have a 3D differentiable rendering pipeline so that the texture could not only be rendered onto a 3D face mask, but also have gradients from the target network backpropagate through it. Zolfi et al. (2022) developed such a pipeline as long as the texture could be fit into a 2D UV mask, shown in figure 2, which we use in DAFR. We find that optimal performance occurs when the texture is resized to fit most of the content within the UV mask, allowing the perturbation to manifest across the majority of the area in the sample.

Generating images that follow text prompts that are also adversarial to facial recognition networks is more abstract (and thus more challenging) than generating samples that use class conditionals of the target network to make it look like another one of the classes – this is without considering the challenges relating to the generated image being a texture applied to a variety of different images, rather than being the final example itself with no concerns for any other image. By optimizing for

multiple distinct images, the aim is so that when the texture is applied to a mask on an unknown image, the mask will be robust to environmental conditions and remain adversarial.

This led to the introduction of multiple mechanisms to control the adversarial elements of generation which emphasize different aspects of texture generation. Firstly, we introduced an inner loop which increased the number of steps of adversarial guidance done per time step (controlled by $k$ in algorithm 1). Having a sufficient number of adversarial steps is important due to the importance of optimizing the texture to work in different conditions. Secondly, we manipulate how early in the reverse process we begin adversarial guidance (controlled by $l$ in algorithm 1) which allows for more adversarial steps, thus better robustness and adversarial strength.

Thirdly, we use DDIM sampling (Song et al., 2021) which allows for a variable number of sampling steps in the time schedule (controlled by $T$ in algorithm 1). Finally, the step size of each adversarial step (controlled by $s$ in algorithm 1) can be changed to influence generation. We find that a constant step size throughout the entire generation leads to adverse perturbations in the later steps of generation, where the noise schedule varies significantly less. We introduce a scaling function in equation (1) to slowly decrease the step size based on the proportion of the time schedule left.

$$f(y) = e^{3(y-0.6)-3\min(0,y-0.5)}, \quad y \in [0,1] \tag{1}$$

Several rounds of tinkering with this equation were required, as initially we used the variance of the noise at each diffusion step, but found that using equation (1) was more effective, potentially due to the later time steps not being scaled to be minuscule.

When DAFR is fully deployed, the final result is shown in figure 1. Compared to previous work (Zolfi et al., 2022; Gong et al., 2024), our masks are significantly stealthier and present new capabilities for these attacks, with respect to the style of generated accessories.

## 3 RESULTS

**Baselines:** To evaluate our accessories, we compare them to recent adversarial face mask attacks. Adversarial Mask (Zolfi et al., 2022), shortened to AdvMask for brevity, generates face masks for dodging by using a 3D differentiable pipeline and optimizing the mask to be adversarial while maintaining a low TV loss. SASMask (Gong et al., 2024) generates face masks for impersonation so that given content is included (e.g., flowers); however, uses a style transfer network to change the style to be optimal (e.g., by changing the colour). AdvMask does not attempt to be faithful to a style so we do not report our stealthiness measure for those masks, while SASMask does so we do for them. We also test a white non-adversarial face mask to act as a non-adversarial baseline for comparison.

**Datasets:** We use two different datasets: PubFig (Kumar et al., 2009), which includes faces of a variety of celebrities, and is where the identities for the dodging benchmark come from, and VGGFACE2-HQ (Chen et al., 2024), which contains GAN upscaled images of the VGGFACE2 dataset (Cao et al., 2018). We randomly choose 100 identities from VGGFACE2-HQ to form part of the finetuned classes and another 900 to be used as part of the threshold selection process.

**Target Networks:** Vakhshiteh et al. (2021) highlight the lack of diversity in the network types studied; therefore, we test on four different network types using different threat models:

1. **Pretrained Large Recognition Models (R100)**: Large pretrained recognition models are often used in previous work (Zolfi et al., 2022) and are publicly available for anyone to use. We test on the pretrained ArcFace ResNet-100[1] directly, that is, before the finetuning in the FT100 setup.

2. **Finetuned Networks (FT100)**: There exists large pretrained backbones that are used for recognition, however, without a head, these networks cannot be used for classification. If a small business wanted to train a recognition network for their employees, then they could do further training on the backbone as well as introducing and training a head. We take a pretrained backbone[1] used in previous work (Zolfi et al., 2022), and perform further training on the 100 identities from VGGFACE2-HQ. This included adding an ArcFace head (Deng et al., 2019a) and training

---

[1]MS1MV3 ResNet-100, available from https://github.com/deepinsight/insightface/blob/master/recognition/arcface_torch/README.md

using the Adam optimizer (Kingma & Ba, 2015) for 100 epochs, while ensuring to use occlusion as an augmentation method during training to improve performance on masked individuals. The final accuracy on 4,500 test images was 97.15%.

3. **Facial Representation Encoder (FaRL)**: We test on the image encoder from FaRL (Zheng et al., 2022), a vision transformer (Dosovitskiy et al., 2021) backbone for face analysis tasks, including recognition. We specifically chose the epoch 16 pretrained backbone, as used by Zheng et al. (2022).

4. **Mobile devices (MFN)**: Mobile devices are common, however, running large networks on them is impossible due to hardware constraints. MobileFaceNet (Chen et al., 2018) is an architecture specifically designed for face recognition and verification on mobile and embedded devices; we test a pretrained MobileFaceNet using weights provided by Sun et al. (2024).

**Threshold Selection:** Previous work has used a mixture of reporting cosine similarities and using success thresholds (Zolfi et al., 2022; Gong et al., 2024; Komkov & Petiushko, 2021). We decide to report both and calculate thresholds using unseen images (i.e., not used elsewhere in the work) from the identities chosen from VGGFACE2-HQ. Previous work (Zolfi et al., 2022; Yin et al., 2021) chooses a threshold that obtains a false acceptance rate (FAR) of 0.01 on masked images

Table 1: Cosine Similarity thresholds and TPRs of the different networks when achieving a FAR of 0.01, the rate of inter-class pairs which are misclassified as intra-pairs. TPR is the proportion of intra-class pairs that are correctly classified as being the same person.

| Network | Class count | Masked | | Unmasked | |
|---|---|---|---|---|---|
| | | *Threshold* | *TPR* | *Threshold* | *TPR* |
| FT100 | 100 | 0.5300 | 0.7835 | 0.0817 | 0.9802 |
| | 1000 | 0.8355 | 0.1799 | 0.8394 | 0.2248 |
| R100 | 100 | 0.2687 | 0.8643 | 0.1788 | 0.9320 |
| | 1000 | 0.2370 | 0.8736 | 0.1757 | 0.9317 |
| FaRL | 100 | 0.7684 | 0.2457 | 0.6670 | 0.4959 |
| | 1000 | 0.7659 | 0.2202 | 0.6568 | 0.4711 |
| MFN | 100 | 0.6156 | 0.3376 | 0.2912 | 0.8114 |
| | 1000 | 0.6622 | 0.2169 | 0.2845 | 0.8212 |

from 1000 identities. Similarly, we use the mean threshold that achieves a FAR of 0.01 over 10 fold cross validation on masked images (with the mask being uniformly chosen between a white, black or blue mask and placed on the face). We calculate separate thresholds for the 100 and 1000 classes chosen from VGGFACE2-HQ. We use the masked thresholds in table 1 throughtout this work, however, for completeness we show the thresholds if unmasked images were used in table 1 as well. Further discussion of the target network's performance is given in appendix A.

**Benchmark Setup:** In the following tests we use our benchmark, FAAB (refer to section 4), to test the dodging capabilities of the different mask generation methods on 30 randomly selected identities from PubFig. Each mask is generated using 25 images of the identity and then tested on 10 other images of that same person. The results are aggregated over all 300 tests and reported. The cosine scores are given in the format: mean $\pm$ standard deviation. Success rate is given by a threshold that is defined as the proportion of tested masked images of the attacker that the embedding of the masked image had a cosine similarity less than the threshold. Each architecture has two thresholds, "SR 100" and "SR 1000" for the 100 and 1000 class thresholds respectively from table 1.

CMMD is performed on the generated texture to measure the stealthiness of an accessory quantitatively (see section 5), but because different attacks use this texture differently in their rendering, we also report CMMD on the final UV 2D mask. Note we use the scaled version of CMMD, with the scale parameters the same as those provided by the authors (Jayasumana et al., 2024).

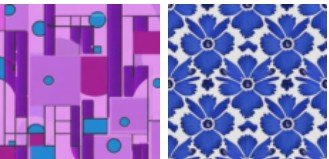

Figure 3: Stealthy masks attempt to be faithful to a reference image. The left image is the reference for purple shapes and the right for blue flowers.

It is important to note that to generate the recognition embedding anchors, we use masked pictures of faces following the procedure of Zolfi et al. (2022) which we now explain. For each identity, 10 unseen images were used for identities from PubFig and 45 images from VGGFACE2-HQ. The mask applied is uniformly chosen from a random noise, white, or black mask. The final anchor embedding is the mean embedding of all the masked images of that identity. Using masked images rather than unmasked images does make the attacks harder, but also prevents the accessory itself from having a impact.

To test the effect of different mask contents/styles, we focus on two different styles for SASMask and DAFR which are advantageous for adversarial masks. An attacker could choose to use any style they want in the real world, so these attacks should be tested as such. The chosen prompts were purple shapes[2] and blue flowers[3]. As SASMask uses images as a content reference, the image produced by the diffusion model using the text prompt is used. Figure 3 shows the reference images.

**Implementation Details:** MTCNN (Zhang et al., 2016) is used for face alignment with R100, FT100, and FaRL, with FFHQ (Karras et al., 2019) face alignment used for MFN. The differentiable 3D mask rendering pipeline from (Zolfi et al., 2022) is used for placing the masks on faces. The hyperparameters of stealthy mask attacks vary the tradeoff between adversarial strength and stealthiness, therefore we test several sets of hyperparameters which balance this tradeoff. We note that for each DAFR attack, we use 200 DDIM sampling steps. We use Stable Diffusion's v2-1 (Rombach et al., 2022) Text to Image LDM[4] as the diffusion model in DAFR.

To indicate the hyperparameters of an attack briefly, we define several abbreviated versions which indicate hyperparameter values (shown in table 2). Different hyperparameters show off not only different tradeoffs between stealthiness and adversarial strength, but are necessary for effective attacks against different networks. AdvMask only has one hyperparameter which is the weight for the TV loss, for which we test two different values.

Table 2: Names and hyperparameter values of different hyperparameter sets for each attack, with DAFR using notation from algorithm 1.

|  | TV Weight | | |
| --- | --- | --- | --- |
| **AdvMask-a** | 0.05 | | |
| **AdvMask-b** | 0.35 | | |
|  | Adv. Weight | | |
| **SASMask-a** | 25 | | |
| **SASMask-b** | 50 | | |
| **SASMask-c** | 600 | | |
| **SASMask-d** | 950 | | |
| **SASMask-e** | 2000 | | |
|  | $l$ | $s$ | $k$ |
| **DAFR-a** | 0.8 | 7 | 5 |
| **DAFR-b** | 0.8 | 7 | 10 |
| **DAFR-c** | 0.8 | 10 | 12 |
| **DAFR-d** | 0.8 | 2 | 1 |

For SASMask, we keep the style hyperparameters consistent across all sets but change the adversarial weight to be the respective number. The style weights then are $\lambda_1 = 1000, \lambda_c = 0.01, \lambda_{tv} = 100, \lambda_s = 10000$ using the notation from the original work (Gong et al., 2024). Note that while the original paper does not have an adversarial weight, the official implementation as of writing has always had one.

**Results:** The results in table 3 show that DAFR is consistently outperforming previous work in stealthiness, reflected by having a lower M-CMMD than previous stealthy mask attacks, which can be visually confirmed in figure 4. In the majority of cases, especially for FT100 in table 3 and MobileFaceNet in table 3, DAFR has managed to accomplish the dual goal of creating adversarial masks that are effective and stealthy. This is due to the adversarial generation process being able to find adversarial textures that do not deviate substantially from the original generation. We believe this is significant progress as no other work to date has managed to preserve the style and content of a given reference image for adversarial accessories as effectively as ours. This can be seen further in the appendices D and E where the style generation capabilities are tested further.

When attacks do not consider stealthiness (such as AdvMask in table 3), then the adversarial strength of the masks is strong, but still not 100% against some of the networks like FT100 and R100. This demonstrates the challenging threat model of adversarial accessories, where critical facial features for recognition can not be manipulated, therefore on unseen images it is difficult to cover every possible transformation. Figure 4 presents some of the masks generated by AdvMask which do not achieve a perfect success rate despite focusing on such.

FT100 was able to achieve the highest TPR on the 100 class problem while achieiving a FAR of 0.01 on unmasked images (table 1). Despite this, FT100 (table 3) is vulnerable to DAFR's capability to produce very stealthy masks while not sacrificing much adversarial strength, compared to AdvMask. This should inform real world decisions to avoid reckless use of such technology as it can appear on the surface to be outstanding while being incredibly vulnerable to adversaries.

Recent work has focused on the architecture of R100 or similar (Zolfi et al., 2022; Gong et al., 2024; Pautov et al., 2019; Komkov & Petiushko, 2021), yet when the same architecture is used but

---

[2]Prompt: "abstract light purple and pink computer pattern with colorful circles, rectangles, triangles and semi circles like it was made in the 1990s"

[3]Prompt: "blue flower pattern"

[4]Weights for the LDM can be found here.

with different weights (such as FT100), the perturbations produced can be drastically different. This is seen in all three attacks tested, suggesting it is not unique to one attack. The masks generated against FaRL also vary dramatically following this trend (see figure 4). One reason this may happen is due to the shape of the gradient output looking like faces, leading to faces being formed in the adversarial generation process with the different attacks handling these faces differently. FT100 has been trained for a specific 100 class recognition problem rather than the thousands of identities trained for in MS1MV3 (Deng et al., 2019b) leading to a weaker separable embedding space which is not discriminative.

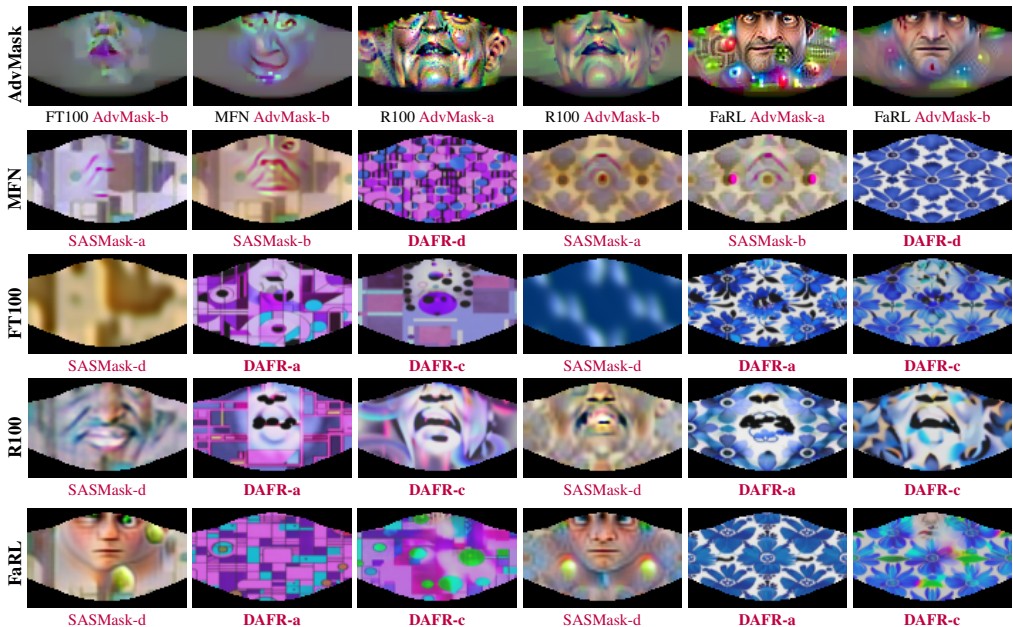

Figure 4: Some of the face mask textures generated as part of the benchmarks within this section for dodging "Beyonce Knowles" in PubFig (Kumar et al., 2009). Underneath each mask is a description of what network it was generated for and the attack used. The top row are all masks generated using the AdvMask attack. For the other rows, the left hand side are generated using the purple shapes style, while the right-hand side are generated using the blue flower style.

# 4 FAAB: FACE ACCESSORY ATTACK BENCHMARK

For an adversarial accessory to be considered robust, it is necessary that it is effective in various environmental conditions including lighting, backgrounds and angles. This is not just an important factor during generation, but also when evaluating an accessory's performance. As the results in section 3 demonstrate, the balance between adversarial strength and stealth has a significant impact on the attack success and so it is crucial to evaluate these factors in order to fairly compare approaches.

To date, there is no standardized framework for testing adversarial face accessories (Vakhshiteh et al., 2021), in part due to the varying threat models and attack objectives for work in the field. Whilst benchmarking frameworks exist for neighboring fields, such as GREAT score (Li et al., 2023) for evaluating general adversarial perturbations using generative models, within the realm of adversarial accessories there are inconsistent experimental frameworks that create hard to compare results. Therefore, we propose a highly adaptable benchmarking framework, titled the Face Accessory Attack Benchmark (FAAB), that is capable of consistent and systematic comparison of different attack methods.

To achieve this feat, FAAB uses a systematic procedure for evaluating accessories, with interchangeable components, detailed below:

- Firstly, an accessory must be generated by the attack being tested for a given individual. It is important at this stage that images used to generate the accessory are not those that will be later used to evaluate, as a strong adversarial accessory should be effective in unseen conditions.

- Once the accessory has been generated, the testing phase begins. This consists of loading in dataset images, placing the accessory on the images using the augmentation method specified by the attack and computing the output of the recognition system.

- Next, we calculate statistics based on adversarial strength and the accessory itself. The statistics calculated are interchangeable and new statistics can be added. By default, we compute the embedding of the backbone network as well as recording the angle of the face within each picture – figure 5 demonstrates the variety of poses tested. Alongside computing statistics based on the attack, we compute statistics on the accessory.

- In order to get a holistic view of the performance of an attack method as a whole, we repeat the above steps over multiple individuals. To further expand analysis, FAAB supports benchmark variations which can modify how images are augmented. For example, we can ensure that the accessory brightness matches that of the image to test different lighting conditions.

- Once all the accessories have been tested, the final stage of the benchmark is to group statistics together based on the properties recorded throughout. This allows us to view the statistics recorded, for example, for each individual, or for images where the individual is looking straight on. How statistics are grouped is entirely customizable and can help discover links between various properties that have been accumulated in the benchmark.

As alluded to above, it is necessary to quantize how stealthy an accessory is as it is infeasible to construct a manner of evaluating stealthiness through the means of a user survey in such a way as to not introduce bias to one attack and to be fair, especially when the definition of stealthiness itself is often subjective. In section 5 we outline why we believe that CMMD is an effective measure of stealthiness and the resulting values are discussed in section 3. Further principles to evaluating style that could be applied to other work are provided in appendix C.

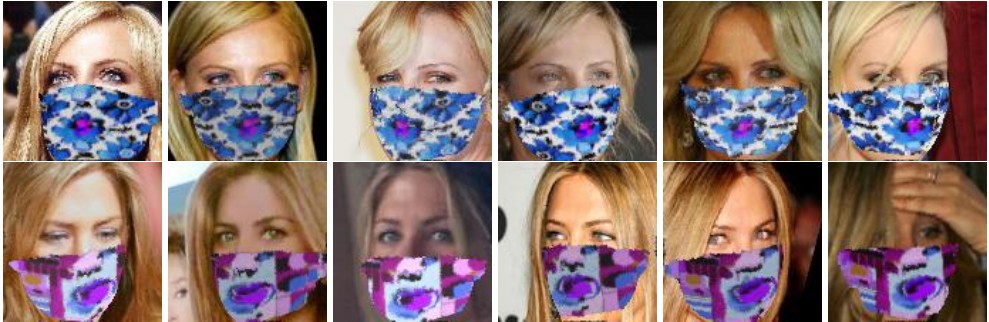

Figure 5: Masked face images of "Charilize Theron" and "Jennifer Aniston" using DAFR-3a in the benchmark in table 3 on FT100. The variety of poses and backgrounds ensures that during generation and evaluation, masks must be robust to real world transforms.

## 5 RELATED WORKS

Here we discuss the closest previous works; we comparatively review further literature across several areas in appendix B.

**Patch-based Adversarial Attacks On Facial Recognition:** Adversarial accessories are small wearables that contain patterns that when placed within an image cause malicious behavior. Previous adversarial accessories have varied significantly in the generation process and in the type of accessory, including glasses (Sharif et al., 2019), hats (Komkov & Petiushko, 2021), face patches (Pautov et al., 2019), eye patches (Xiao et al., 2021) and face masks (Zolfi et al., 2022; Gong et al., 2024). As mentioned in section 1, face masks have seen an increase in usage within the general public and are a prime adversarial accessory as they cover a substantial area of the face (Zolfi et al., 2022; Gong et al., 2024), hence they are our chosen accessory type.

Most adversarial accessory attacks primarily focus on the accessory being adversarial (Sharif et al., 2019; Pautov et al., 2019; Komkov & Petiushko, 2021) or focus on emphasizing additional properties such as transferability (Xiao et al., 2021; Gong et al., 2024; Zolfi et al., 2022). However, the generated accessories from these works do not look like "normal" attire and would arouse suspicion if worn in the real world – we would consider these to not be stealthy. Gong et al. (2024) attempts to explicitly generate stealthy face masks using adaptive styles and style losses, but these cause the texture to depart from a reference image significantly. We focus primarily on stealthiness and argue that for a mask to be stealthy, it must look similar to a reference image. To the best of our knowledge, we are the first work to utilize diffusion models to generate adversarial accessories.

**Quantitative Measures Of Style:**  Within adversarial attacks on facial recognition, some style measures that have been used before in makeup attacks (Sun et al., 2024) include SSIM (Wang et al., 2004), PSNR and FID (Heusel et al., 2017). Gong et al. (2024) used SSIM in a setup specific to their face mask which is hard to transfer to other attacks. Creating metrics to evaluate the quality of generated images is a problem faced by generative models and stealthy adversarial accessories that generate textures can be seen as generators that have a baseline style (a "real" set) which can generate multiple adversarial textures (a "generated" set). CLIP Maximum Mean Discrepancy (CMMD) (Jayasumana et al., 2024) is a recent metric proposed to measure the quality of generated images by finding the maximum mean discrepancy (MMD) (Gretton et al., 2006; 2012) between CLIP (Radford et al., 2021) embeddings of a real and generated set of images. An unbiased estimator of MMD on two sets of CLIP embeddings, $X = \{x_1, x_2, ..., x_m\}$ and $Y = \{y_1, y_2, ..., y_n\}$, and kernel $k$ (for which we use a RBF kernel) can be given by the equation below. For the results in this paper, we scaled the output of CMMD for display purposes, using the same values as in the original paper (Jayasumana et al., 2024).

$$dist^2_{MMD}(X,Y) = \frac{1}{m(m-1)} \sum_{i=1}^{m} \sum_{j \neq i}^{m} k(x_i, x_j) + \frac{1}{n(n-1)} \sum_{i=1}^{n} \sum_{j \neq i}^{n} k(y_i, y_j) - \frac{2}{mn} \sum_{i=1}^{m} \sum_{j=1}^{n} k(x_i, y_j)$$

By using CLIP embeddings, CMMD is able to provide a more holistic evaluation of style and has been found to outperform FID and other common metrics when compared to human raters (Jayasumana et al., 2024) which resonates back to the user surveys in previous accessory work (Sharif et al., 2019). We believe future work should also use CMMD as a metric to evaluate the quality (thus stealthiness) of their generated textures and so having an evaluation that is similar to how image generators are evaluated. More details about how we use CMMD are found in section 4.

## 6 CONCLUSION

We propose a novel diffusion-based attack, DAFR, for adversarial mask generation that generates masks that are both adversarial and stealthy. We demonstrate the effectiveness of the attack on a range of architectures and threat models, and highlight the challenges in attacking these models. Moreover, we propose a robust standardized benchmarking framework, FAAB, for evaluating the strength and stealthiness of these attacks such that comparisons between future work can be quicker, robust, and fair. This further invites future work to use this framework to create strong and stealthy adversarial accessories.

**Limitations:**  We have tested different attacks on a variety of networks and have found that the behavior of the attacks can vary significantly between networks. This unfortunately means that DAFR can struggle to produce stealthy adversarial masks on the strongest networks. Additionally, DAFR uses adversarial guidance (Dai et al., 2023) and is sensitive to the hyperparameters highlighted in table 2. Small changes can lead to significant variation in the output and their values must be adjusted for different target networks. This requires manual testing to balance the stealthiness of the generated mask and its stealthiness.

**Future Work:**  Generating stealthy masks on stronger networks is difficult and future work could expand the number of networks these masks are stealthy for. Finally, all adversarial face mask attacks are inherently vulnerable to removal by generative based defenses. Given the weights of a face mask removal network, future work could generate masks that are adversarial to the recognition network and the removal network to mitigate this weakness.

**FT100, finetuned backbone**

| Attack | Style | Cosine Sim. (↓) | SR 100 (↑) | SR 1000 (↑) | T-CMMD (↓) | M-CMMD (↓) |
|---|---|---|---|---|---|---|
| Non Adv. | White | $0.6701 \pm 0.2280$ | 0.2767 | 0.7083 | / | / |
| AdvMask-a | Random | $0.0363 \pm 0.2490$ | 0.9467 | 0.9833 | / | / |
| AdvMask-b | | $-0.011 \pm 0.1856$ | 0.9983 | 0.9983 | / | / |
| SASMask-d | Purple Shapes | $0.3893 \pm 0.3341$ | 0.6227 | 0.8933 | 3.1881 | 1.7600 |
| SASMask-e | | $0.4568 \pm 0.3278$ | 0.5133 | 0.8550 | 3.1208 | 2.0857 |
| DAFR-a | | $0.2288 \pm 0.2451$ | 0.8553 | 0.9733 | **1.3295** | **0.8471** |
| DAFR-b | | $0.2166 \pm 0.2411$ | 0.8867 | 0.9683 | 1.6960 | 1.1771 |
| DAFR-c | | $\mathbf{0.1858 \pm 0.2232}$ | **0.9133** | **0.9767** | 2.3490 | 1.5018 |
| SASMask-d | Blue Flowers | $0.4167 \pm 0.3699$ | 0.5733 | 0.8383 | 3.8306 | 3.5290 |
| SASMask-e | | $0.3728 \pm 0.3384$ | 0.6500 | 0.9000 | 4.0924 | 2.9544 |
| DAFR-a | | $0.2455 \pm 0.2384$ | 0.8633 | 0.9717 | **2.4527** | **1.0090** |
| DAFR-b | | $0.2253 \pm 0.2316$ | 0.8850 | **0.9783** | 2.8263 | 1.1199 |
| DAFR-c | | $\mathbf{0.1752 \pm 0.2213}$ | **0.9167** | 0.9750 | 3.6985 | 1.6210 |

**MFN, MobileFaceNet**

| Attack | Style | Cosine Sim. (↓) | SR 100 (↑) | SR 1000 (↑) | T-CMMD (↓) | M-CMMD (↓) |
|---|---|---|---|---|---|---|
| Non Adv. | White | $0.7006 \pm 0.0477$ | 0.0500 | 0.1500 | / | / |
| AdvMask-a | Random | $0.3051 \pm 0.0861$ | 1.0000 | 1.0000 | / | / |
| AdvMask-b | | $0.3502 \pm 0.0922$ | 1.0000 | 1.0000 | / | / |
| SASMask-a | Purple Shapes | $0.3932 \pm 0.1740$ | 0.9233 | 0.8783 | 2.3938 | 1.8368 |
| SASMask-b | | $\mathbf{0.1272 \pm 0.1155}$ | **1.0000** | **1.0000** | 3.2128 | 2.2126 |
| DAFR-d | | $0.2952 \pm 0.1312$ | **1.0000** | **1.0000** | **1.2792** | **0.8416** |
| SASMask-a | Blue Flowers | $0.2701 \pm 0.0921$ | 1.0000 | 1.0000 | 3.9781 | 3.0724 |
| SASMask-b | | $\mathbf{0.1661 \pm 0.0996}$ | **1.0000** | **1.0000** | 4.1066 | 2.9479 |
| DAFR-d | | $0.4805 \pm 0.0899$ | 0.9417 | 0.9850 | **0.9009** | **0.1040** |

**R100, pretrained backbone**

| Attack | Style | Cosine Sim. (↓) | SR 100 (↑) | SR 1000 (↑) | T-CMMD (↓) | M-CMMD (↓) |
|---|---|---|---|---|---|---|
| Non Adv. | White | $0.6141 \pm 0.1981$ | 0.0783 | 0.0783 | / | / |
| AdvMask-a | Random | $0.031 \pm 0.1313$ | 0.9467 | 0.9183 | / | / |
| AdvMask-b | | $0.0341 \pm 0.1291$ | 0.9567 | 0.9216 | / | / |
| SASMask-c | Purple Shapes | $\mathbf{0.1096 \pm 0.1280}$ | **0.8900** | **0.8233** | 3.0740 | 2.5177 |
| SASMask-d | | $0.1118 \pm 0.1392$ | 0.8883 | 0.8200 | 3.3451 | 2.5361 |
| SASMask-e | | $0.1164 \pm 0.1584$ | 0.8600 | 0.8050 | 3.4989 | 2.9703 |
| DAFR-a | | $0.2523 \pm 0.1513$ | 0.5750 | 0.4733 | **1.5728** | **1.2138** |
| DAFR-b | | $0.1988 \pm 0.1579$ | 0.6867 | 0.6316 | 2.2289 | 1.7903 |
| DAFR-c | | $0.1963 \pm 0.1633$ | 0.7183 | 0.6350 | 3.0438 | 2.4713 |
| SASMask-c | Blue Flowers | $0.1066 \pm 0.1325$ | 0.8800 | 0.8266 | 4.1449 | 4.5054 |
| SASMask-d | | $0.0921 \pm 0.1371$ | 0.9200 | **0.8700** | 4.4332 | 4.7600 |
| SASMask-e | | $\mathbf{0.0854 \pm 0.1285}$ | **0.9250** | 0.8667 | 4.5162 | 3.4040 |
| DAFR-a | | $0.2670 \pm 0.1471$ | 0.2432 | 0.4367 | **2.9012** | **1.2724** |
| DAFR-b | | $0.2052 \pm 0.1501$ | 0.7033 | 0.6317 | 3.9275 | 1.7545 |
| DAFR-c | | $0.1924 \pm 0.1604$ | 0.7100 | 0.6383 | 4.9682 | 2.4842 |

**FaRL, pretrained ViT**

| Attack | Style | Cosine Sim. (↓) | SR 100 (↑) | SR 1000 (↑) | T-CMMD (↓) | M-CMMD (↓) |
|---|---|---|---|---|---|---|
| Non Adv. | White | $0.8053 \pm 0.0553$ | 0.1917 | 0.2017 | / | / |
| AdvMask-a | Random | $0.3848 \pm 0.1017$ | 1.0000 | 1.0000 | / | / |
| AdvMask-b | | $0.4037 \pm 0.1069$ | 1.0000 | 1.0000 | / | / |
| SASMask-d | Purple Shapes | $\mathbf{0.4364 \pm 0.0988}$ | **1.0000** | **1.0000** | 2.7931 | 2.6759 |
| SASMask-e | | $0.4450 \pm 0.1223$ | **1.0000** | **1.0000** | 2.7984 | 2.5965 |
| DAFR-a | | $0.7471 \pm 0.0524$ | 0.6100 | 0.6283 | 1.5283 | 0.8031 |
| DAFR-b | | $0.7327 \pm 0.0530$ | 0.7200 | 0.7350 | **1.2023** | **0.9583** |
| DAFR-c | | $0.7159 \pm 0.0546$ | 0.8367 | 0.8533 | 1.4417 | 0.8877 |
| SASMask-d | Blue Flowers | $\mathbf{0.4006 \pm 0.0957}$ | **1.0000** | **1.0000** | 4.9293 | 3.4139 |
| SASMask-e | | $0.4047 \pm 0.0981$ | 0.9983 | 0.9983 | 4.9995 | 4.8929 |
| DAFR-a | | $0.7663 \pm 0.0495$ | 0.4367 | 0.4700 | **1.7892** | **0.5895** |
| DAFR-b | | $0.7576 \pm 0.0501$ | 0.5100 | 0.5400 | 2.4341 | 1.0099 |
| DAFR-c | | $0.7254 \pm 0.0541$ | 0.7933 | 0.8117 | 3.0446 | 1.1998 |

Table 3: Results of the four dodging benchmarks of the different networks tested, as outlined in section 3. The first set of columns indicate the attack and its style, the second set of columns indicate the attack statistics aggregated over the 300 test images in each benchmark when the targeted attacker wears the mask, and the final set of columns are accessory statistics aggregated over the 30 generated textures.

Arrows next to each column indicate the desired direction of each metric, for example ↓ would indicate lower values are desirable. Cosine similarity is in the format of *Mean ± Std-Deviation* over the test images. SR 100 and SR 1000 are the success rate of the dodging masks over the test set using the thresholds in table 1. T-CMMD and M-CMMD are defined as the CMMD (refer to section 5) over the texture images and mask texture images (see figure 2 for the difference). Different attacks convert their texture onto the mask differently therefore M-CMMD is a fairer evaluation. For each column within attacks using the same style, a marker has been used to indicate rank: **1st**, *2nd* and 3rd.

DAFR outperforms SASMask for every network in terms of stealth (shown in the CMMD columns), while either outperforming SASMask adversarially or by sacrificing minimal adversarial strength.

## 7 ETHICS STATEMENT

Deep learning based facial recognition and verification systems are becoming more prominent around the world. On one hand, DAFR highlights new security risks to existing face recognition and verification systems by creating masks that are indistinguishable from more colorful masks people wear; which could undermine their efficacy and the trust put in them. However, by demonstrating these capabilities, future defenses and adversarial training schemes will have to consider these types of accessories thus allowing future work to defend against DAFR or a more advanced version of it. On the other hand, these powerful systems can be misused by different institutions and the existence of these accessories demonstrate that these systems are not flawless and can be manipulated in certain circumstances.

## 8 REPRODUCIBILITY STATEMENT

All the work for this project was performed on a single NVIDIA A5000 GPU. Depending on the attack type and hyperparameters, each benchmark could take between 40 minutes to 7 hours to generate all 30 different adversarial textures used for results in section 3. To evaluate the different metrics evaluated within these benchmarks on the GPU would require around 30 minutes. The main body contains 50 benchmarks which would take roughly 286 hours on a single GPU, with the appendix benchmarks taking a further 100 hours. The supplementary material contains all the code to run the work, including Python code for all the attacks, benchmark and other utilities (such as threshold selection etc.). Instructions have been provided to help run the code.

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

## A  FURTHER DETAILS ON RESULT SETUP

Table 1 demonstrates the performance of the different target networks. Pretrained backbones have been trained to have a highly discriminative embedding space across a wide range of datasets, rather than a separable one across one dataset. This leads them to perform incredibly well on unseen faces and to have the highest TPR in table 1. FT100 was trained for the 100 class scenario and therefore performs well, but then struggles on 1000 classes. This threat model has not been explicitly explored before (with previous work performing further training on their models (Gong et al., 2024)) and highlights vulnerabilities to these models if deployed recklessly. FaRL is a general purpose face encoding and so has not been explicitly trained for recognition, explaining the lower TPR. On the other hand, MFN has been trained for recognition but we expect its smaller size may limit its performance. Despite this, MFN performs the best out of all the networks tested at not being fooled by the non-adversarial mask when using the 100 class threshold and second best when using the 1000 class threshold, demonstrating that it is still an effective network, shown in table 3.

## B  EXTENDED RELATED WORKS

**Facial Recognition:**  Facial recognition systems have evolved significantly over the last couple of decades, with the state of the art approaches using deep learning models that are able to achieve high accuracy in both large and small class sizes. Traditionally, for a small number of classes in a closed-set environment (that is the test set consists only of identities from within the training set), softmax-based approaches that are used in general object recognition can be effective. However, softmax losses encourage the learned features to be separable, but not necessarily discriminative (Wen et al., 2016), leading to worse performance when there are lots of classes of data or in an open-set environment, where the test set includes identities not in the training set (Liu et al., 2017).

Focus has moved to using Siamese networks (Bromley et al., 1993) where the backbone learns a discriminative, rather than separable, embedding space through different losses such as center loss (Wen et al., 2016) or triplet loss (Schroff et al., 2015). More recent work has focused on maximizing the angular margins of learnt class centers in a learnt embedding space, such that embeddings from a given class' center point are in a similar direction and that embeddings not from that center's class, point in a different direction (Liu et al., 2017). Several works have aimed for intra-class compactness and inter-class discrepancy with the aim of learning a discriminative embedding space (Liu et al., 2017; Wang et al., 2018; Deng et al., 2019a).

**Adversarial Examples Using Generative Models:**  Traditionally, adversarial examples were generated using gradient based methods such that the perturbation has a small matrix norm; one example is using projected gradient descent (PGD) (Madry et al., 2018). However, Song et al. (2018) used generative models (specifically generative adversarial networks) to construct unrestricted adversarial examples that exhibit greater realism. This has progressed so that recently there have been several works proposing diffusion model based adversarial attacks using different techniques (Xue et al., 2023; Chen et al., 2023; Dai et al., 2023). Using diffusion models for this task has several benefits, most notably, greater controllability and visual fidelity of generated samples (Dai et al., 2023), areas in which previous adversarial accessories struggled. Our attack, therefore, leverages these properties through the use of a textually controlled diffusion model.

**Patch-based Adversarial Attacks And Defenses:**  Adversarial patches (Brown et al., 2017) are small patterns that when placed within an input image, cause unintentional behavior in a network. To improve the robustness of these patches to real world conditions, past work has shown that it is necessary to incorporate real world transformations into the generation process (Athalye et al., 2018). In the context of adversarial accessories, this translates to generating on different images of a person, for example at different poses, such that different transformations are considered during accessory generation.

From the perspective of defending adversarial attacks, recent defenses have utilized diffusion models (Nie et al., 2022) for purification of adversarial examples, removing the perturbation while maintaining the original content. For adversarial patches, these techniques have been found to be inadequate, therefore, specific adversarial patch defenses have been developed (Kang et al., 2023). These defenses use diffusion models to locate the patch and then replace it using inpainting, which could be

used to replace a face mask with an estimated face. Another defense would be to remove face masks from images using generative models trained to do so (Kumar et al., 2023). All current adversarial face masks are vulnerable to these last two defenses and so creating robustness to these defenses is not within the scope of our work and could be the goal of future work.

**Other Attacks On Facial Recognition:** There have also been other attacks on these systems such as adversarial makeup (Yin et al., 2021) which has been used so that the attack can access a wider area of the face rather than just a local patch. Recent work has used diffusion models to enhance this approach further (Sun et al., 2024) creating highly realistic makeup to fool these models. These attacks have a significantly larger area of the face to attack and may be difficult to physically realize compared to face masks.

Another channel of attack is using visible light (Shen et al., 2019; Nguyen et al., 2020; Li et al., 2020) where perturbations are projected onto the face. These attacks offer a different representation to their perturbations which presents unique challenges which could also be explored with diffusion models in a similar fashion to our work.

Backdoor attacks on face recognition have also been developed, where the system behaves as expected on clean input but then has been modified to behave erroneously on malicious input. These attacks can be split into three components: the attack channel (the attacker's knowledge and access to the victim model), the injection method (how the manipulation can occur) and the trigger method (what triggers the corruption) (Roux et al., 2024). One such example work has directly manipulated weights such that only certain identities are misclassified, but the rest are unaffected (Zehavi & Shamir, 2023).

Another type of attack are those that poison the training set of a victim model such that when the attacker wears a physical accessory, the model erroneously classifies them (Chen et al., 2017).

These methods have a different threat model to our work, but are a different avenue of work that could be expanded using diffusion models.

## C EVALUATING STYLES

**Previous Style Metrics:** As previously discussed, most adversarial accessory work has not focused on stealth and so there is a limited range of quantitative measures for stealthiness. Previous work has used TV loss to ensure accessories are color smooth, making them easier to physically realize and less noise like, but often stealthiness is not explicitly measured after generation (Zolfi et al., 2022; Komkov & Petiushko, 2021; Pautov et al., 2019).

Stealthiness is a subjective measure so an ideal method would be to collect user surveys, as has been done before (Sharif et al., 2019) where participants were asked to identity whether given images of glasses were "real" or generated. While this does gather valuable user opinion, these surveys are time consuming to run, potentially hard to reproduce when ran on a small scale and may not accurately measure stealthiness (as the concept is abstract to the general public). Some measures that have been used before in makeup attacks (Sun et al., 2024) are SSIM (Wang et al., 2004), PSNR and FID (Heusel et al., 2017). In recent stealthy mask work, Gong et al. (2024) measure the SSIM of masked faces with the mask texture being the original pattern in the style of their adversarial pattern and then comparing these images to masked faces with the adversarial pattern. Whilst these measures are able to yield valuable statistics about a generated accessory, we believe these do not capture the true essence of stealthiness – a better metric would be one which determines the quality of generated images. This can be achieved by treating the adversarial attack as an image generator and using similar metrics to measure its performance such as CMMD.

**Proper Use Of CMMD:** When choosing images for CMMD evaluation for general accessory evaluation in future work, we recommend trying to evaluate on as close of a representation as the texture in the final accessory while avoiding any faces being in the images (such as the images in figure 4). Furthermore, the reference set for the style should be one image representing the style the textures in the generated set are attempting to create. The generated set should contain multiple textures from different attacks (i.e different attackers/targets) of the same style.

## D  TESTING DIFFERENT STYLES

Section 3 focused on two styles that were chosen due to being effective for the adversarial mask generation. However, to demonstrate the effectiveness of the stealth based approaches on a wide range of styles we test both DAFR and SASMask on 20 randomly chosen text prompts from a filtered set of DrawBench prompts (Saharia et al., 2022). Stealthy approaches may try to "hide" their perturbations in the content making more abstract content better as the content can vary significantly while still being faithful. Prompts from DrawBench are more concrete and contain a wide range of content, and test whether these attacks can still be stealthy even when not given an advantageous style. The same dodging benchmark was used as has been used in section 3, but 5 identities were chosen out of the previous 30 with the same number of images used for generation and testing as used previously.

| Attack | Style | SR 100 (↑) | SR 1000 (↑) | Mask CMMD (↓) |
|---|---|---|---|---|
| SASMask-b | Blue Dog | 1.0 | 1.0 | 3.7062 |
| DAFR-d | | 1.0 | 1.0 | 2.5122 |
| SASMask-b | Panda Emoji | 1.0 | 1.0 | 2.2752 |
| DAFR-d | | 1.0 | 1.0 | 0.5084 |
| SASMask-b | Black Sandwich | 1.0 | 1.0 | 5.9267 |
| DAFR-d | | 0.99 | 0.99 | 1.3971 |
| SASMask-b | Owl | 1.0 | 1.0 | 3.9005 |
| DAFR-d | | 1.0 | 1.0 | 0.7783 |
| SASMask-b | Giraffe | 1.0 | 1.0 | 0.6479 |
| DAFR-d | | 1.0 | 1.0 | 0.6479 |
| SASMask-b | Bricks | 1.0 | 1.0 | 4.2328 |
| DAFR-d | | 1.0 | 1.0 | 1.3618 |
| SASMask-b | **Overall** | 1.0 | 1.0 | 3.7150 |
| DAFR-d | | 0.9925 | 0.998 | 1.4791 |

Table 4: Some of the results from the style attack test on MobileFaceNet, for 6 out of the 20 styles chosen from filtered DrawBench. These tests use the same metrics as table 3, so a smaller success rate (SR 100 and SR 1000) is desirable.

Table 4 shows the results of our test on MobileFaceNet. Both attacks are successfully able to fool the network consistently, however DAFR generates stealthier masks as demonstrated by CMMD and by the visual results shown in figure 6. DAFR can achieve adversarial strength by manipulating the content of the textures in a manner faithful to the style, such as changing the hat, eyes and mouth of the panda in figure 6. We expand this study to FT100 and R100 in appendix E.

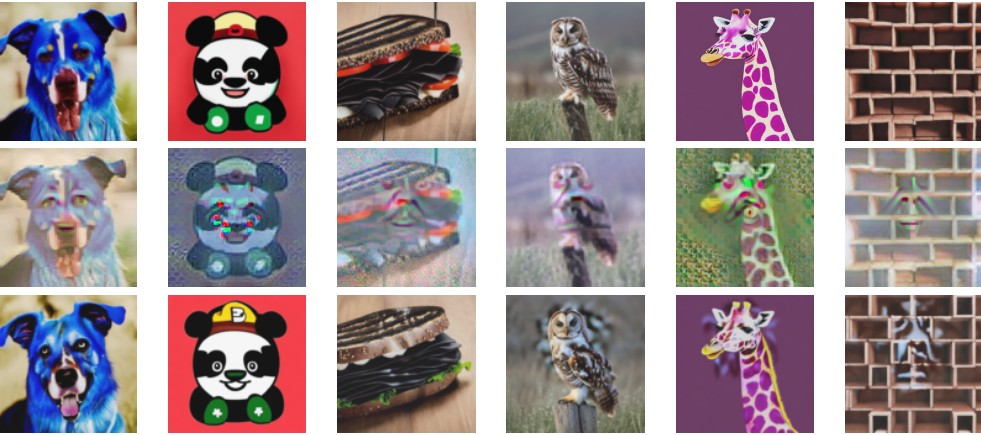

Figure 6: Textures from masks trying to dodge from the "Kiera Knightley" identity from the style test. The top row is reference images, the next row is generated by SASMask-b and the final row is generated by DAFR-d.

## E   EXPANDING THE STYLE TEST

We conduct the same study as performed in appendix D, but using FT100 and R100. The attacks were less successful against these networks (refer to table 3) so these tests demonstrate the attacks ability to remain stealthy in a more difficult scenario.

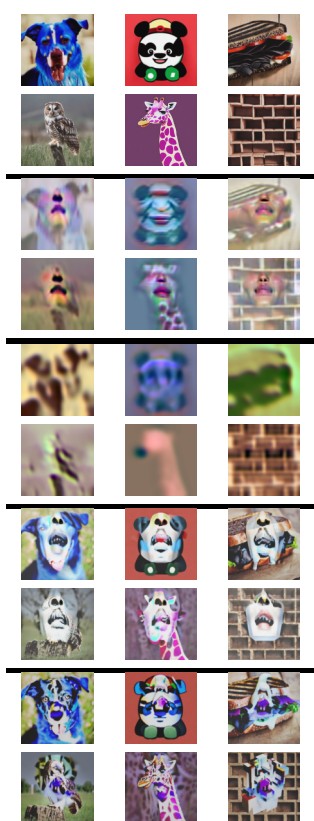

| Attack | Arch. | Style | Cosine ($\downarrow$) | M-CMMD ($\downarrow$) |
|---|---|---|---|---|
| SASMask-d | R100 | Blue Dog | 0.1434 | 5.4164 |
| DAFR-b | | | 0.1920 | 2.6407 |
| SASMask-d | R100 | Panda Emoji | 0.1167 | 4.9877 |
| DAFR-b | | | 0.2322 | 3.7407 |
| SASMask-d | R100 | Black Sandwich | 0.0802 | 4.8970 |
| DAFR-b | | | 0.1998 | 2.6640 |
| SASMask-d | R100 | Owl | 0.1318 | 7.1038 |
| DAFR-b | | | 0.1946 | 5.2481 |
| SASMask-d | R100 | Giraffe | 0.1057 | 3.5506 |
| DAFR-b | | | 0.2267 | 1.514 |
| SASMask-d | R100 | Bricks | 0.1068 | 4.5495 |
| DAFR-b | | | 0.1895 | 2.6462 |
| SASMask-d | R100 | **Overall** | 0.1366 | 4.9009 |
| DAFR-b | | | 0.1962 | 2.8633 |
| SASMask-d | FT100 | Blue Dog | 0.2172 | 6.1107 |
| DAFR-b | | | 0.1711 | 3.3500 |
| SASMask-d | FT100 | Panda Emoji | 0.1558 | 5.3880 |
| DAFR-b | | | 0.2505 | 4.4208 |
| SASMask-d | FT100 | Black Sandwich | 0.5254 | 5.1856 |
| DAFR-b | | | 0.2041 | 2.4467 |
| SASMask-d | FT100 | Owl | 0.3621 | 6.0591 |
| DAFR-b | | | 0.3057 | 3.4554 |
| SASMask-d | FT100 | Giraffe | 0.3684 | 4.3279 |
| DAFR-b | | | 0.2837 | 1.9312 |
| SASMask-d | FT100 | Bricks | 0.2904 | 3.9697 |
| DAFR-b | | | 0.2188 | 2.8751 |
| SASMask-d | FT100 | **Overall** | 0.3610 | 4.8306 |
| DAFR-b | | | 0.1989 | 2.6857 |

Table 5: A table showing some of the results from the style attack test using 6 out of the 20 styles chosen from filtered DrawBench. Cosine is the mean cosine, while M-CMMD is the masked texture CMMD from table 3.

Figure 7: Grid of textures from masks trying to dodge from the "Kiera Knightley" identity from the style test. The sections from top to bottom are reference images, SASMask-d R100, SASMask-d FT100, DAFR-b R100, DAFR-b FT100

Table 5 and figure 7 display selected results and textures from the style test on FT100 and R100. Firstly, DAFR outperformed SASMask on these obscure styles both stylewise and adversarially on FT100, while performing slightly worse adversarially on R100. Both attacks have significantly higher mask CMMD values compared to the tests with an advantageous style in previous sections. While an attacker can always choose to use an advantageous style, future work should focus on making an attack that can work on a wider range of styles.

## F   ANGLE STATISTICS

An advantage of using FAAB is that a deeper understanding of the different properties of an accessory is evaluated such as its robustness to different face poses, with figure 5 demonstrating the variety of poses. We now analyze the effectiveness of the different face mask attacks when they are used at different angles. To measure the pose of each face, the yaw, pitch and roll are calculated, allowing the images to be classified into two categories: straight on and angled. Straight on images represented around 67% of the images while angled represented 31% of the images with the remaining 2% representing images with an extreme yaw and patch.

1. Straight on images have the magnitudes of yaw and pitch less than 15 degrees.

2. Angled images have either their yaw or pitch with a magnitude greater than 15 degrees while still both having a magnitude less than 45 degrees.

| Architecture | Attack | Straight On | | Angled | |
|---|---|---|---|---|---|
| | | *SR 100* (↑) | *SR 1000* (↑) | *SR 100* (↑) | *SR 1000* (↑) |
| FT100 | Non Adv | 0.2450 | 0.6900 | 0.2978 | 0.7660 |
| | AdvMask-b | 0.9975 | 0.9975 | 0.9894 | 1.0000 |
| | SASMask-d | 0.5850 | 0.8300 | 0.5426 | 0.8457 |
| | DAFR-b | 0.9125 | 0.9825 | 0.8404 | 0.9680 |
| R100 | Non Adv | 0.0525 | 0.0525 | 0.0744 | 0.0744 |
| | AdvMask-b | 0.9725 | 0.9400 | 0.9202 | 0.8776 |
| | SASMask-d | 0.9450 | 0.8950 | 0.8617 | 0.8085 |
| | DAFR-b | 0.7225 | 0.6500 | 0.6436 | 0.5691 |
| FaRL | Non Adv | 0.16 | 0.1725 | 0.1649 | 0.1809 |
| | AdvMask-b | 1.0000 | 1.0000 | 1.0000 | 1.0000 |
| | SASMask-d | 1.0000 | 1.0000 | 1.0000 | 1.0000 |
| | DAFR-b | 0.4775 | 0.5075 | 0.4775 | 0.5075 |
| MFN | Non Adv | 0.0570 | 0.1373 | 0.1520 | 0.3039 |
| | AdvMask-b | 1.0000 | 1.0000 | 1.0000 | 1.0000 |
| | SASMask-b | 1.0000 | 1.0000 | 1.0000 | 1.0000 |
| | DAFR-d | 0.9197 | 0.9819 | 0.9804 | 0.9901 |

Table 6: A table containing the results of the attacks at different angles. When the attack has a style, we show the blue flower pattern style. These results come from benchmarks from the earlier sections or identical reran benchmarks.

Table 6 shows the efficacy of the different masks when tested on different angles. It is important to notice that all the non adversarial masks become more effective when the attacker is not straight on. However, when it comes to adversarial face masks, the performance tends to decrease (such as on FT100, R100), which may occur due to the mask being an effective attack in most cases and when the mask is not in the image, the benefit of being angled is less than the benefit of the adversarial texture itself. Despite this, on FaRL the angled version has a negligible impact on performance and improved DAFR's performance on MFN showing that this phenomena is not universal.