# OpenReview forum: "Text To Stealthy Adversarial Face Masks"
_ICLR.cc/2025/Conference — ICLR 2025 Conference Withdrawn Submission_

### Official Review · Reviewer_aYrd · 2024-10-31

**Soundness:** 2
**Presentation:** 1
**Contribution:** 2
**Rating:** 3
**Confidence:** 5

**Summary:**

The authors propose Diffusion Attack against Facial Recognition (DAFR), a method capable of generating robust and stealthy face masks for dodging recognition systems using the guidance of textual prompts to determine the style. A new benchmark is also presented, the Face Accessory Attack Benchmark (FAAB) which includes a set of standardized tests and procedures to evaluate the performance of accessories.

**Strengths:**

1. The literature review is thorough and effectively encompasses key works relevant to the field.

**Weaknesses:**

1. Motivation: While I recognize the significance of prior works highlighting face masks as a potential threat vector during the COVID-19 pandemic, their prevalence has markedly declined since then. As a result, I find it challenging to accept the notion that face masks, irrespective of printed patterns, can currently be considered a genuinely stealthy accessory.
2. Writing quality: the submission is mainly held back by the writing quality and lack of clarity in most sections. These are mainly focused around the method section (Section 2), which seems a bit unprofessional, lacking formulaic descriptions of relevant preliminaries and the attack’s full pipeline (e.g., how the components interact with each other) and overall extremely unorganized. I suggest the authors reorganize (e.g., split the section into subsections, each describing the main steps of the attack) and improve (formalize the entire pipeline) this section for better clarity of the novelties they propose.
3. Novelty: The proposed method appears to lack substantial originality, as it primarily combines existing approaches adapted to this specific task, without introducing significant additional contributions. For instance, the mask rendering technique for facial images is adopted from Zolfi et al. (2022), while the diffusion-based adversarial attack approach is drawn from Dai et al. (2023), with adjustments made for the face recognition domain. Although the authors mention the use of textual prompts to control style, the method section lacks a clear methodological explanation of this aspect, including details on how prompts are selected and their impact on the attack's objectives.
4. Evaluation: While the authors assess their attack against various baselines and across multiple models and datasets, several critical aspects remain unaddressed:
- Shallow Analysis – The authors predominantly present empirical results without delving into deeper insights, examining edge cases, or discussing unexpected findings.
- Missing Ablation Studies – For instance, Equation 1 claims that a scaling function is superior to a static value, yet no comparative analysis is provided to validate this assertion.
- Lack of Transferability Experiments – A crucial aspect of adversarial attacks is their transferability to models beyond those on which they were trained. Testing transferability could offer valuable insights into the practicality of the proposed attack.
- Absence of Real-World Experiments – Although digital evaluations form the core of the paper, an attack using a practical accessory would benefit from real-world testing to demonstrate efficacy beyond digital scenarios.
- Implementation Details – The section includes an excess of low-level details (specific steps for each decision), which detracts from the key information. I recommend prioritizing content between the main paper and the appendix, allowing for more space for additional experiments, such as those in Sections D, E, and F of the appendix.
- Results – The results across most models do not consistently outperform the baselines (notably SASMask). Ideally, the proposed method should exhibit at least comparable performance to baselines while enhancing stealthiness. The current configuration seems to prioritize stealthiness at the expense of attack success.

Minor comments:
1. Line 89 – CMMD is not introduced until line 446 (not even a reference to it).
2. Line 125 – unclear why $f$ is mentioned here.
3. Line 137 – unclear what $R$ is.
4. Line 301 – table 3 is too far from where it was mentioned in the text. Maybe the table could be split to better fit in the flow of the text.

**Questions:**

Please see the weaknesses above.

---

### Official Review · Reviewer_NidS · 2024-11-03

**Soundness:** 2
**Presentation:** 2
**Contribution:** 2
**Rating:** 3
**Confidence:** 4

**Summary:**

This paper proposes to use diffusion models to generate adversarial accessories (a mask) for attacking facial recognition. Two important attack properties, i.e., robustness (resilient to changes in viewing angles and environmental conditions) and stealthiness (do not attract suspicion by, for example, incorporating obvious facial features), are considered.

**Strengths:**

1. Applying text-to-image diffusion models to optimizing adversarial accessories is new.
2. Experiments are conducted on multiple datasets, models, text-guided styles, and with various metrics.
3. The proposed attack outperforms the baselines regarding stealthiness.
4. The attempt to form a benchmark is promising.

**Weaknesses:**

- This paper has claimed to improve the attack in both robustness and stealthiness. However, the evaluations in the main body are limited to the stealthiness (and the attack strength in the common 2D digital setting). Although some robustness results are added to the Appendix (Table 6), those results show that the proposed attack is worse than the baselines. In addition, no physical, 3D experiments are considered.
- As acknowledged by the authors, the idea of using diffusion models to generate adversarial examples is not new. However, it is not clearly stated what the specific (technical) challenge is for generating adversarial masks, compared to other forms of perturbations, such as the makeups. Without identifying such challenges, the technical novelty of this paper is not clear.
- It is interesting to report the results for different text prompts. However, since different prompts lead to dramatically different attack performance (see Table 4), it would be necessary to understand the relation between the prompt and the performance and finally learn to select the optimal one.
- Presentation issues:
This paper contains lots of technical details but lacks an overview of the whole attack pipeline (maybe in the form of a flowchart).
Before introducing their method, the paper should include foundational knowledge related to adversarial masks and clarify the meanings of the mathematical symbols. For instance, the meanings of x and I in Algorithm 1 are not defined.

**Questions:**

How does DAFR perform in the physical world (3D robustness) and against black-box models (attack transferability)?

---

### Official Review · Reviewer_QP84 · 2024-11-03

**Soundness:** 3
**Presentation:** 3
**Contribution:** 2
**Rating:** 3
**Confidence:** 4

**Summary:**

The paper proposes a method to generate adversarial face masks that can fool face recognition systems in a white box setting. They specifically borrow the adversarial attack framework from AdvDiff.

**Strengths:**

It is an interesting work on a relevant problem. The method performs well over previous baselines and including 3D rendering bridges the gap towards real world applicability.

**Weaknesses:**

1. The attack is a simple adaptation of previous approaches i.e. AdvDiff.
2. What is R in the Algorithm 1? Should it be $h_i$ instead of $h_n$?
3. The generation process is firstly done in a white-box setting, while this in itself is not problematic the authors have not included any results on transferability to see whether it can be used for another model?
4. The attack is not robust against state-of-the-art facial recognition models such as ArcFace. The stealth to attack success rate trade-off for R100 is quite large. SASMask achieves almost the same T-CMMD and M-CMMD scores with significantly higher SR.
5. Line 208: Which dataset was the ArcFace model trained on?
6. The test set consisting of 300 images is not statistically significant.
7. A white mask has a SR 1000 of 0.7083 against the fine-tuned F100 model. Is this scenario even worth studying? This simply means that the model does not perform well to begin with.
8. Is the attack success impacted by the facial pose?

**Questions:**

See weaknesses section.

---

### Official Review · Reviewer_9QYn · 2024-11-04

**Soundness:** 1
**Presentation:** 2
**Contribution:** 2
**Rating:** 3
**Confidence:** 4

**Summary:**

This paper focuses on adversarial attacks on face recognition models using face masks, claiming that existing methods lack in attack stealthiness. It proposes a diffusion-based adversarial face mask generation method, titled DAFR. DAFR controls the generation of the diffusion model using adversarial guidance. Additionally, this paper builds a benchmark to evaluate the performance of existing adversarial accessories.

**Strengths:**

1. This paper addresses an important issue, namely the security of face recognition models, which is meaningful for both academia and industry.
2. The paper leverages the ability of diffusion models to generate realistic and natural images to create adversarial face masks, which is an meaningful design.

**Weaknesses:**

1. Lack of real-world attack experiments. This paper focuses on the stealthiness of adversarial attacks; however, discussing stealthiness without considering real-world attack implementation seems meaningless. For instance, digital adversarial attack methods can use subtle perturbations undetectable to the human eye, achieving stealthy attacks. The reason existing methods lack stealthiness is due to the need for higher perturbation intensity to achieve real-world attacks. Discussing attack stealthiness without addressing real-world implementation is thus unconvincing.
2. Evaluation of stealthiness. Stealthiness is a subjective evaluation dimension, and it is unclear if the quantitative metric CMMD used in this paper matches human perception. I suggest adding a user study to support the paper's claims of improved stealthiness.
3. Lack of ablation experiments. There is insufficient analysis of the effectiveness of key components. For example, to improve robustness, the authors optimize the adversarial pattern on a set of images of the attacker, 𝐻; however, it remains unclear if this design actually improves robustness.
4. Missing essential references: Towards Effective Adversarial Textured 3D Meshes on Physical Face Recognition, CVPR 2023

**Questions:**

1. It is recommended that the authors create an adversarial mask to test the attack effectiveness of DAFR in the real world.
2. It is suggested that the authors add an ablation study of 𝐻: conduct an experiment comparing performance with and without optimizing over multiple images of the attacker.
3. Does text prompt have an effect on the results?
4. This paper should include a framework diagram to illustrate the DAFR method, which would enhance readability.

---

### Note · Authors · 2024-11-13

**Comment:**

We thank the reviewers for their work.

**Withdrawal Confirmation:**

I have read and agree with the venue's withdrawal policy on behalf of myself and my co-authors.